# Discrimination in Online Markets: Effects of Social Bias on Learning from Reviews and Policy Design

**Faidra Monachou**
Stanford University
monachou@stanford.edu

**Itai Ashlagi**
Stanford University
iashlagi@stanford.edu

## Abstract

The increasing popularity of online two-sided markets such as ride-sharing, accommodation and freelance labor platforms, goes hand in hand with new socioeconomic challenges. One major issue remains the existence of bias and discrimination against certain social groups. We study this problem using a two-sided large market model with employers and workers mediated by a platform. Employers who seek to hire workers face uncertainty about a candidate worker's skill level. Therefore, they base their hiring decision on learning from past reviews about an *individual* worker as well as on their (possibly misspecified) prior beliefs about the ability level of the *social group* the worker belongs to. Drawing upon the social learning literature with bounded rationality and limited information, uncertainty combined with social bias leads to unequal hiring opportunities between workers of different social groups. Although the effect of social bias decreases as the number of reviews increases (consistent with empirical findings), minority workers still receive lower expected payoffs. Finally, we consider a simple directed matching policy (DM), which combines learning and matching to make better matching decisions for minority workers. Under this policy, there exists a steady-state equilibrium, in which DM reduces the discrimination gap.

## 1 Introduction

Online markets such as ride-sharing, accommodation, and freelance labor platforms have grown rapidly over the past few years, thus shaping the future of work. However, a major issue, which is common in traditional markets, is the existence of bias and discrimination against certain social groups. Indeed, several empirical studies document the existence of racial, gender and other forms of discrimination in popular online platforms. In experiments on Airbnb, an online accommodation-sharing platform, Edelman et al. [18] and Cui et al. [16] find that accommodation applications from guests with distinctively African-American names are about 16%-19% less likely to be accepted relative to identical guests with distinctively white-sounding names. A study by Ge et al. [25] confirms analogous results for race discrimination on the ridesharing platform Uber while Ameri et al. [5] document discrimination against travelers with disabilities on Airbnb. Hannák et al. [27] examine racial and gender discrimination in two freelance labor markets, TaskRabbit and Fiverr; on both platforms, workers perceived to be black get worse ratings than similarly qualified workers perceived to be white while on TaskRabbit women receive fewer reviews than men with equivalent work experience.

Understanding the effects of social bias on learning from reviews will help designing successful interventions that reduce the existing discrimination. Towards this goal, we consider a two-sided large market model of employers and workers mediated by platform. Workers belong to one of

two different social groups (minority or majority[1]) and can be either high-skilled or low-skilled; employers may or may not be biased against minority workers. Employers are matched randomly with candidate workers and decide whether to hire them or not, but, due to the uncertainty about a worker's skill level, base their decision (i) on past reviews about the individual worker, (ii) on their private prior beliefs about the skill level of the social group that the worker belongs to, and (iii) on personal preferences. Employers in the model may be biased against the minority group of workers. We study the dynamics of employers' beliefs under social bias. Although social bias decreases with additional reviews, the welfare of minority workers is lower than majority workers at the steady-state equilibrium of the market (Theorem 1). We also design a simple algorithmic policy to decrease discrimination; our proposed DM (Directed Matching) policy uses a combination of *learning* and *matching* [30] to make better matching decisions for minority workers and improve welfare. DM learns (within a small error probability) the group of an employer and then matches employers to majority or minority workers based on the employer's identified group. The policy aims at protecting minority workers from matching with discriminating employers and results in Pareto improvement over the benchmark uniformly random matching (UM) algorithm that ignores social bias (Theorem 3). The DM policy can explain ubiquitous policies that boost new workers to the top of search results in online labor platforms.

The behavioral assumptions in this paper are motivated by the rich empirical and theoretical literature which has identified two potential sources of discrimination: *belief-based* (mostly known as statistical) and *taste-based*. Regarding the former, a long line of research in the statistical discrimination literature assumes that group differences do exist and are exogenous [22]. Thus, employers hold correct beliefs about aggregate group differences (e.g. [6, 36, 4, 13, 28, 34]). This case is not considered here since we assume an equally distributed skill level between the two social groups of workers. Instead, we consider an alternative source of belief-based bias that recent research has demonstrated: *incorrect prior beliefs* [11, 39, 12, 24, 23, 16]. Without being aware of their own bias (see [37] about the bias blindspot effect), some employers may hold misspecified models of group differences which, in the absence of perfect information, lead to false judgment of an individual's abilities. The combination of both uncertainty and *misspecification* results in discrimination.

The key difference between belief-based and taste-based social bias is the effect of information. Taste-based discrimination models [8] assume that the differential treatment of minority groups is driven purely by preferences; thus, discrimination persists even with perfect information about an individual's true skills. In sharp contrast to taste-based bias, the presence of more information gradually reduces (and asymptotically eliminates) the effect of belief-based bias. The existing empirical research on online platforms indicates the existence of belief-based bias. For example, Cui et al. [16] find that a positive review can significantly alleviate racial discrimination on Airbnb while self-claimed quality information by guests themselves cannot. Using Airbnb observational data, Abrahao et al. [1] also find that users with higher reputation scores are considered more trustworthy regardless of their demographic characteristics. Finally, several papers such as [38] and [35] suggest that rating systems can be utilized towards combating discrimination in online markets. Motivated by these findings, we focus mainly on belief-based bias and show that the model is consistent with the empirical findings (Theorem 2); nevertheless, we also discuss taste-based bias (Supplementary Material, Section E.1).

From a technical point of view, we draw upon a variety of tools to prove our results. First, employers engage in a naïve social learning process from reviews to learn about the true quality of each worker in the market. In contrast to the existing literature [40, 10, 7, 9, 41, 15, 29, 2, 17, 26, 20, 19, 21, 33], we also assume that employers may have different (misspecified) priors. We use a stochastic approximation analysis [9] to represent the dynamics of the social learning process. Second, we adopt a continuum model and extend the setting in [30] by including agent histories on both sides of the market as well as by incorporating a social learning component with employer incentives (in their hiring and review decisions) to the evolution of the system. These two factors, along with a different objective and the presence of social bias, make the dynamical system in our paper take a non-linear form, and differentiate (both technically and conceptually) our model and policy from [30]. Finally, regarding the learning algorithm used in our DM policy, we also use results by Agrawal et al. [3] on a variation of the stochastic multi-armed bandit (MAB) problem.

## 2 Model

We consider an online labor platform of *workers* and *employers*[2] mediated by a platform. Time is discrete $t = 1, \ldots$ and a mass of workers and employers arrive at each period $t$. We assume that each arriving worker and employer stays for $K$ periods; later, we also consider the limit $K \to \infty$. The market is initially empty, and no workers and employers have entered the market before time $t = 1$. At each time $t$, each worker is paired with one prospective employer who decides whether to hire the worker or not. Throughout the paper, we use $t$ to denote the absolute time in the market, $k = 1, \ldots, K$ to denote the relevant period during a worker's lifetime and $n = 1, \ldots, K$ to denote the relevant period during an employer's lifetime.

**Agents.** At any time $t$, a mass $\lambda_B > 0$ of *minority* ($B$) and a mass $\lambda_A > 0$ of *majority* ($A$) workers arrives to the market. Regardless of his social group $c \in \{A, B\}$, each worker may be *high*-skilled or *low*-skilled; let $Q \in \{H, L\}$ denote the skill level of the worker and let the true fraction of high-skilled workers within a social group $c$ be $q_0 \in (0, 1)$. The social group of each worker is publicly observed by the platform and the agents, but only nature knows the true skill level of a worker.

Each employer belongs to one of two groups $e \in \{N, D\}$ based on her prior belief: *non-discriminating* ($N$) employers and *discriminating* ($D$) employers against minority workers. At each time period $t$, a mass $\lambda_N > 0$ of $N$ employers and a mass $\lambda_D > 0$ of $D$ employers arrive to the market.

**Matching, Actions and Utility.** At each period $t$, employers and workers who are still present in the market are paired by the platform; initially we assume uniformly random pairing. For simplicity, we assume that workers always accept the incoming offers of employers. If hired at one period, the worker receives payoff 1 for that period, otherwise he gets zero payoff. Each employer has two actions at each period; she can either *hire* ($m = 1$) or *reject* ($m = 0$) the candidate worker. By slightly abusing notation, we call employer $k$ the $k$-th employer that a certain worker meets over his lifetime. If employer $k$ hires the worker, she receives utility

$$U_k = A_k + \mathbf{1}_{\{Q=H\}} + P_k, \tag{1}$$

where $A_k$ is an ex ante idiosyncratic term and $P_k$ is an ex post[3] idiosyncratic term. Otherwise, she receives zero utility. The random variables $A_k$ and $P_k$ are independent; they are also i.i.d. with known continuous CDF $F_A$ and $F_P$ and bounded support $[\underline{a}, \overline{a}]$ and $[\underline{p}, \overline{p}]$, respectively. Let $\mu_P = \mathbb{E} P_k$. Finally, note that the exact values of $A_k$ and $P_k$ are privately revealed only to the $k$-th employer of the worker.[4]

**Rating system.** Given a positive hiring decision, there is a probability $\eta > 0$ that the employer leaves a review. A review $r_k$ can be either *good* ($g$) or *bad* ($b$) while $r_k = \diamond$ denotes that no review was left. Reviews are imperfect as they are determined by the realized utility $U_k$ of each employer $k$. In particular, $r_k = g$ if $U_k \geq 0$ and $r_k = b$ if $U_k < 0$. Upon meeting a candidate worker, employer $k$ observes the worker's social group $c$, her private value $A_k$, and some information on worker's history provided by the platform. The platform observes the full history of a worker (consisting of hiring decisions and reviews) but the employers see only the statistics of good and bad reviews. Specifically, let $G_k$ and $B_k$ denote the number of good and bad reviews *before* period $k$. Hence, in any period $k$, employers only observe $G_k$ and $B_k$. Before the worker enters the market ($k = 1$), no reviews are available thus $B_1 = G_1 = 0$.

**Belief-based social bias.** The platform has the correct prior belief $q_0$ about minority and majority workers' skill level. All employers also share the same, correct prior belief $q_0 \triangleq \frac{G_0}{N_0} \in (0, 1)$ about majority workers. Regarding minority workers, non-discriminating ($N$) employers have prior belief $\frac{G_0^N}{N_0} = q_0 = \frac{G_0}{N_0}$ but discriminating employers ($D$) use a *misspecified* prior belief $\frac{G_0^D}{N_0} = \beta q_0 = \frac{\beta G_0}{N_0}$. In this case, we say that discriminating employers have *social bias level* $\beta \in (0, 1)$ against minority workers.

**Naïve learning from reviews.** We consider that employers have limited computational ability and naively use the fraction of good reviews (adjusted by their prior belief)

$$q_k^e = \frac{G_k + G_0^e}{G_k + B_k + N_0} \tag{2}$$

as a proxy for the probability that the worker is high-skilled. The fictitious reviews $G_0, N_0 \in \mathbb{N}$ may be interpreted as the weight that employers assign to the prior belief $q_0$ (see (2)). The smaller the number of reviews, the more employers rely on their private belief. Hence, as the number of reviews increases, employers start relying more on the external information (see also Lemma 2). If the total number of reviews $N_k \triangleq G_k + B_k = 0$, $q_k^e$ is equal to employer $k$'s prior belief $\frac{G_0^e}{N_0}$.

**Worker welfare.** At the steady-state equilibrium of the market[5], we define the *worker's welfare* as

$$W_Q^c(K) = \mathbb{E}\left( \sum_{k=1}^{K} \delta^k m_k \mid Q, c \right), \tag{3}$$

where $\delta \in (0, 1)$ is a known discount factor and $m_k$ is the hiring decision of the $k$-th employer.

We say that there exists *discrimination* against minority (resp. majority) workers if $W_Q^B(K) < W_Q^A(K)$ (resp. $W_Q^A(K) < W_Q^B(K)$) for all $Q \in \{H, L\}$. For life time $K$, we define the *discrimination gap* $d(Q, K)$ among workers of skill level $Q$ as $d(Q, K) = W_Q^A(K) - W_Q^B(K)$.

**Assumptions.** We make the following two technical assumptions.

**Assumption 1 (Richness)** *The support $[\underline{a}, \overline{a}]$ and $[\underline{p}, \overline{p}]$ of random variables $A$ and $P$ are such that*

$$\mathbb{P}(A + \mu_P \geq 0) = \mathbb{P}(A \geq -\mu_P) > 0, \tag{4}$$

*i.e. $\overline{a} + \mu_P \geq 0$, as well as $\underline{a} + \mu_P + q_0 \leq 0$. Furthermore,*

$$\underline{a} + \overline{p} > 0 \ \text{ and } \ \overline{a} + 1 + \underline{p} < 0. \tag{5}$$

**Assumption 2 (Balanced market)** *The market is perfectly balanced, i.e. $\lambda_D + \lambda_N = \lambda_A + \lambda_B$.*

Assumption 1 is important in establishing almost sure convergence. Specifically, $\overline{a} + \mu_P \geq 0$ ensures that, regardless of the current belief $q_k^e$ of group $e$ employers, there is always a positive probability that the worker is hired; $\underline{a} + \mu_P + q_0 \leq 0$ implies that for belief $q_0$ or smaller, there is a positive probability that the worker is not hired. Finally, (5) guarantees that, conditional on hiring, good and bad reviews happen with positive probability. However, we also examine the case where Assumption 1 does not hold. For simplicity, we also assume that the market is balanced (Assumption 2). This implies that neither employer nor worker stay unmatched in any period. We discuss the case of unbalanced markets in Section 5.

## 3 Effects of Belief-Based Social Bias

In this section, we analyze the dynamics of belief updating during the lifetime $k = 1, \ldots, K$ of a worker in the market. The worker of social group $c$ and unknown (but fixed) skill level $Q$ meets one employer per period $k$. By comparing the case of workers with different social groups but same skill level, we study how social bias affects worker welfare at the steady-state equilibrium of the market as well as asymptotic learning of worker quality.

**Employer's hiring decision problem.** At period $k$, the candidate worker's past reviews $G_k$ and $B_k$ coupled with employer $k$'s prior about group $c \in \{A, B\}$ induce a *belief* $q_k^e$, $e \in \{N, D\}$, regarding his skill level. Upon meeting the candidate worker, employer $k$'s decision problem is simply given by

$$m_k = \arg \max_{m \in \{0,1\}} \mathbf{1}\{m = 1\}(A_k + q_k^e + \mu_P). \tag{6}$$

In turn, the employer accepts the current worker if and only if her expected utility for that worker is non-negative, that is $q_k^e \geq -A_k - \mu_P$.

As a warm-up, we prove the following intuitive properties. Their proofs are straightforward and can be found in the Supplementary Material (together with all other proofs).

**Lemma 1** *The difference $q_k^N - q_k^D$ between the beliefs of group $N$ and group $D$ employers about the same minority worker is positive for any period $k$ and weakly decreases with $k$.*

The following lemma is an immediate corollary.

**Lemma 2** *Fix period $k$ and review statistics $G_k$, $B_k$. Under uniformly random matching, the probability that a minority worker is hired at period $k$ equals*

$$\frac{\lambda_D}{\lambda_D + \lambda_N}(1 - F_A(-q_k^D - \mu_P)) + \frac{\lambda_N}{\lambda_D + \lambda_N}(1 - F_A(-q_k^N - \mu_P)), \qquad (7)$$

*which is smaller than the probability $1 - F_A(-q_k^N - \mu_P)$ that a majority worker with the same review statistics is hired at period $k$.*

In practice, Lemma 1 and Lemma 2 suggest that the *difference* in hiring probabilities of minority and majority workers is large for workers with few reviews.

**Discrimination and worker inequality.** Under the social learning dynamics that we described here, the large market - which we model as a discrete-time dynamical system - always reaches a unique steady state equilibrium (see Lemma A.1 in Appendix A). In the following theorem, we quantify the effect of social bias on the worker welfare at the steady-state equilibrium, and show the existence of discrimination against minority workers.

**Theorem 1 (Discrimination under belief-based social bias)** *At the steady-state equilibrium of the market, there exists discrimination against minority workers, i.e. minority workers have lower expected welfare $W_Q^B(K) < W_Q^A(K)$ than majority workers of the same skill level $Q \in \{H, L\}$.*

**Asymptotic learning under belief-based social bias.** In the baseline model, we have assumed that workers (and employers) stay for a limited time of $K$ periods. Next we verify that learning occurs in the limit $K \to \infty$. Given the "naive" learning rule in (2), we prove that the skill level estimate $q_t^e$ of employer group $e$ about the skill level of a worker asymptotically converges to an estimate $q_\infty^e$. Similar results are also established in [9] and [41]; the main difference lies at the existence of employer groups with contradicting prior beliefs in the case of minority workers.[6] Theorem 2 below shows that the asymptotic estimates for minority workers do not differ between discriminating and non-discriminating employers. It also shows that, despite the naiveté of the social learning rule and employers' different prior beliefs, the employers correctly estimate a higher skill-level for high-skilled workers compared to their estimated level for low-skilled workers. Hence, they are able to distinguish between high-skilled and low-skilled workers.

**Theorem 2 (Asymptotic Learning under Belief-Based Social Bias)** *Fix a worker of social group $c \in \{A, B\}$ and true skill level $Q \in \{H, L\}$. Then, as $K \to \infty$,*

$$q_t^e \to q_\infty^e(Q) \text{ almost surely.}$$

*The limit $q_\infty^e(Q)$ depends on the true skill level $Q$ of the worker but not on his social group $c$, i.e. $q_\infty^N(Q) = q_\infty^D(Q) \equiv q_\infty(Q)$ for workers of the same $Q$. Specifically, $q_\infty(Q)$ is the unique solution to*

$$\mathbb{P}[A + \mathbf{1}_{\{Q=H\}} + P \geq 0 \mid A + q + \mu_P \geq 0, Q] - q = 0, \qquad (8)$$

*which does not depend on the employers' prior beliefs.*

*Furthermore, employers are able to distinguish between high-skilled and low-skilled workers, i.e. $q_\infty(H) > q_\infty(L)$.*

The proof can be found in Appendix B. An important technical observation is that, when employers do not hire the worker or do not leave a review, $N_k = G_k + B_k$ remains unchanged. Formally, let $\tau_j$ denote the time indices of employers who hire the worker and leave a review, i.e. $\tau_1 = \min(k \mid m_k = 1, r_k \neq \diamond)$ and $\tau_j = \min(j \mid j > \tau_{j-1}, m_j = 1, r_j \neq \diamond)$. Since $\tau_{N_k} < k$ denotes the last time a review was left before period $k$, the belief $q_t^e$ has the same value in the time between periods $\tau_{N_k} + 1$ and $\tau_{N_k+1}$, i.e.

$$q_k^e = q_{\tau_{N_k}+1}^e \qquad (9)$$

for any group $e \in \{N, D\}$.

Therefore, for the case of majority workers where both $D$ and $N$ employers share the same belief $q_k$ at any time $k$, the dynamics of $q_k$ at times $\tau_1 + 1, \ldots, \tau_{N_k} + 1$ can be described by a stochastic approximation (Robbins-Monro) algorithm. A known result (Lemma D.2) about the almost sure convergence of *Robbins-Monro* algorithms guarantees the almost sure convergence of $q_t$. On the other hand, the case of minority workers is more complicated. Employers of group $D$ and $N$ co-exist in the market. Due to their different priors, the beliefs of $D$ and $N$ employers do not follow the same dynamics although the probability of getting hired at time $t$ depends on both $q_t^D$ and $q_t^N$. By using a generalized Robbins-Monro argument, we prove that $q_t^D$ and $q_t^N$ converge almost surely to the same limit as the belief about majority workers. Intuitively, as time $t$ grows, Lemma 1 implies that the difference between $q_t^N$ and $q_t^D$ decreases thus both $N$ and $D$ employers gradually forget their prior beliefs and behave in a similar way.

Given the context of belief-based discrimination, the results in Theorem 2 are not surprising. Belief-based discrimination occurs because (possibly misspecified) prior beliefs about *group* characteristics fill in the gap of perfect information about an *individual* worker. Already in Lemma 1, we have shown that the difference $q_k^N - q_k^D$ decreases as the number of reviews increases; thus, if it was possible to collect an infinite number of reviews about a certain worker, uncertainty would eventually disappear and employers would be able to perfectly distinguish between workers of high and low skill level. Despite this fact, note that, for $\delta \in (0, 1)$, the discrimination gap in the worker welfare (as shown in Theorem 1) would still exist in the limit $K \to \infty$: minority workers eventually have equal hiring opportunities but this is not enough to account for the initial social bias in the first periods of their lifetime. Hence, discrimination (in terms of discounted total welfare) persists even as $K \to \infty$.

Finally, we examine the case where Assumption 1 does not hold and find that with positive probability hirings stop and convergence to limit $q_\infty(Q)$ does not occur. Specifically, we have that[7]:

**Lemma 3** *Fix $Q \in \{H, L\}$. Suppose that condition (4) of Assumption 1 does not hold, i.e. $\overline{a} + \mu_P < 0$. Then, with positive probability $q_t^e$ does not converge to $q_\infty(Q)$.*

Practically speaking, the presence of a worker in a real online market, could be indeed very short. Regarding online labor markets, Hannák et al. [27] find that the perceived gender and race have significant negative correlations with search rank and that the number of completed tasks is positively correlated with the number of reviews. Consequently, it may be possible that minority workers may not even have the chance to receive enough reviews or even stay long in the platform due to the tough competition. Lemma 3 partially captures this situation showing that learning stops with positive probability. Nevertheless, considerations about the effect of market congestion (due to competition) and algorithmic bias on the exit rate of minority workers are out of the scope of the current paper but are definitely an interesting direction for future research.

## 4 Policy Design

Under a uniform matching (UM) policy, a minority worker's welfare is smaller than a majority worker's welfare (Theorem 1). We are interested in designing a matching policy that reduces the discrimination gap $d(Q, K)$ for each skill level $Q$ so that it Pareto-dominates uniform matching in terms of worker welfare. A combination of employer type learning and improved matching is used to achieve this goal.

### 4.1 Learning employer types

Given a known social learning model and the history of employers' hiring decisions, the platform can learn - within a reasonable error probability - the group that each employer belongs to. Intuitively, if the platform observes that a certain employer rejects minority workers more often than a non-discriminating employer would do, then this employer probably belongs to group $D$.

**Preliminaries.** We introduce the following definitions which can also be found in Appendix A.1. Let $\Omega_k$, $k = 1, \ldots, K$ denote the worker history of length $k - 1$. Formally, the *worker history* $\Omega_k = \{\omega_1, \ldots, \omega_{k-1}\}$ consists of all the past hiring decisions $m_k \in \{0, 1\}$ and reviews $r_k \in \{\diamond, g, b\}$

for that worker, that is $\omega_k = (m_k, r_k)$. Let $H_n = \{h_1, \dots, h_{n-1}\}$, $n = 1, \dots, K$, also denote the *employer history* of length $n - 1$. Each $h_n$ consists of the hiring decision $m_n \in \{0, 1\}$ made by that employer about a worker with history $\Omega_k$[8] and social group $c \in \{A, B\}$, i.e. where $h_n = (m_n, \Omega_k, c)$. Initially, $H_1 = \emptyset$ and $\Omega_1 = \emptyset$.

Given the employer's history $H_n$ of length $n - 1$, we define $l_n$ to be the *log-likelihood ratio*

$$l_n = \log \frac{\mathbb{P}(g = D \mid H_n)}{\mathbb{P}(g = N \mid H_n)}, \tag{10}$$

where $l_1 = \frac{\lambda_D}{\lambda_N}$. Given thresholds $\theta_N, \theta_D > 0$, an employer is said to be *D-identified* if $l_n > \theta_D$ and *N-identified* if $l_n < -\theta_N$. We also say that the employers who have not been identified yet but match to minority workers are in the *learning pool*. Hence, the learning pool consists of all the minority workers and the unidentified employers matched to them.

**Learning in finite expected time.** For simplicity, suppose that we use a common threshold $\theta$ for both types $N$ and $D$. It turns out that, even for a very large threshold $\theta$, we can manage to learn the type of each employer in expected finite time during their lifetime. On top of that, the high threshold $\theta$ also provides a very high accuracy to the types assigned to employers. To avoid introducing additional notation, we provide below an informal version of this technical result. The proof is based on an application of Lemma 4.3 in [3] (Lemma D.5 in Appendix).

**Lemma 4 (Informal - Lemma C.1 in Appendix C)** *Suppose that a fixed employer is paired to a minority worker $n$ according to a known distribution of worker histories, i.i.d. for each period $n = 1, \dots, K$ of the employer's lifetime. Then, for large enough $K$ and $\theta$, the expected time until an employer of group $e \in \{N, D\}$ gets e-identified is at most $K$, i.e.*

$$\mathbb{E}(\inf\{n > 0 : l_n \geq \theta\} \mid e = D) < K \text{ and } \mathbb{E}(\inf\{n > 0 : l_n \leq -\theta\} \mid e = N) < K. \tag{11}$$

## 4.2 Directed Matching (DM) policy

The platform can use the information provided by the learning algorithm in various ways. This paper offers a simple directed matching (DM) policy that takes advantage of the previous learning algorithm in order to reduce the discrimination gap between minority and majority workers. The policy extends ideas found in [30] and works as follows. In the learning pool, the platform learns the type of each employer by observing her past decisions about minority workers. To protect the minority workers, this particular employer who has been identified as being discriminating should not be matched with minority workers as long as the capacity constraints under arrival rates $\lambda_i$, $i \in \{A, B, N, D\}$ allow to. Therefore, as soon as a mass of employers exits the learning pool, an equal mass of minority workers is matched with employers who have been identified as non-discriminating. On the other hand, $D$-identified employers are matched to majority workers. The idea is simple but, as Theorem 3 shows, it can reduce the discrimination gap.[9]

At the time $t = T_{\text{DM}}$, when the DM policy is introduced to the market, the market is at the steady state equilibrium under the initial uniform matching (UM) policy (see Lemma A.1). Then, DM proceeds as follows for some thresholds $\theta_N$ and $\theta_D$.

At each time $t = T_{\text{DM}}, \dots$ repeat:

1. **Learning.**

   1. Check each employer of history length $n$, for all $n = 1, \dots, K$:
      (a) If $l_n > \theta_D$, identify the employer as $D$.
      (b) If $l_n < -\theta_N$, identify the employer as $N$.
      (c) Otherwise, the employer remains in the learning pool.

   2. All the employers who have just been identified as $N$ or $D$ exit the learning pool.

2. **Matching.**

1. Match the mass of $D$-identified employers to an equal mass of majority workers (uniformly at random).

2. Match the mass of $N$-identified employers to an equal mass of minority workers (uniformly at random). Prioritize over workers who have already been matched with $N$-identified employers in the past. If necessary, select (uniformly at random) a mass of minority workers to exit the learning pool and match them with the remaining $N$-identified employers.

3. Uniformly at random select a mass of newly arrived employers to replace an mass of the employers who have exited either the learning pool or the market so that the total mass of employers and workers in the learning pool are equal. If the workers in the learning pool outnumber the employers in the learning pool, add an appropriate mass of non-identified employers to the learning pool (selected uniformly at random).

4. Match the minority workers in the learning pool to an equal mass of employers in the learning pool (uniformly at random).

5. In any of the previous steps, match any remaining unmatched workers and employers uniformly at random.

**Theorem 3** *For large enough $K$, $\theta_N$ and $\theta_D$, there exists a steady-state equilibrium such that the DM policy Pareto-dominates the UM policy. That is, $d_{\mathrm{DM}}(Q, K) < d_{\mathrm{UM}}(Q, K)$ while $W^A_{Q,\mathrm{UM}}(K) = W^A_{Q,\mathrm{DM}}(K)$ and $W^B_{Q,\mathrm{UM}}(K) < W^B_{Q,\mathrm{DM}}(K)$.*

The intuition behind Theorem 3 is as follows. At the steady-state equilibrium of the market, an incoming minority worker either enters the learning pool (step 2c) or gets matched with $N$-identified employers (step 2b) or matches uniformly at random with employers as in UM (step 2e). For large enough appropriately chosen $\theta_N$ and $\theta_D$, the fraction of $D$ employers in the learning pool at every time $t$ remains at most $\frac{\lambda_D}{\lambda_D + \lambda_N}$. The error probability of the learning algorithm also becomes negligible (Lemma C.3 in Supplementary Material), meaning that almost all $N$-identified employers are indeed $N$ employers. Hence, any incoming minority worker matches in expectation to fewer $D$ employers than he did under the UM policy. This leads to improvement of minority workers' welfare in total. Majority workers are not affected by the employer type and thus earn the same expected welfare.

Finally, observe that Theorem 3 is silent about employers' welfare. An interesting question is whether one can design a matching policy, which in addition to decreasing the discrimination gap, does not harm workers and non-discriminating employers.

## 5  Conclusion and Open Questions

The framework studied in this paper, albeit stylized, provides simple insights about the underlying discrimination mechanism in online two-sided markets. We assumed a behavioral model of naïve agents with belief-based bias. However, other behavioral assumptions are interesting to study. Except for Bayesian agents and taste-based bias, one can assume agents update their (potentially misspecified) beliefs based on their own past experiences. Furthermore, the proposed DM policy is not a panacea but demonstrates useful insights towards eliminating discrimination. The policy exploits the online nature of platforms and the plethora of available data in order to protect against discrimination. Thus, it becomes an example of how platforms can exercise their control towards a less discriminatory environment.

Identifying discriminating employers could also be useful for other platform interventions. The platform could send warnings to or even ban $D$-identified users.[10] Learning possibly discriminating users could also be useful when the platform designs information campaigns that target those users. If discrimination is belief-based, such a measure might be very effective at helping people correct their misspecified prior beliefs. Nevertheless, legal constraints and ethical considerations should also be taken into account since there is no available law that clearly regulates discrimination in platforms [35, 38]. Note that, given our framework, discrimination occurs because in expectation $D$ employers reject minority workers *more often* than they reject majority workers. This does not necessarily imply

that the platform is able to immediately take legal action against the $D$-identified users. In this case, alternative operational measures have to be taken.

This paper offers some insights about discrimination, but also sheds light on even bigger questions that remain open for future research. Reviews play an important role in many decisions that online platforms make. The rating system creates an information loop: the platform relies on user-generated data to learn about workers while employers' decisions and feedback are partially based on information provided by the platform. For example, the platform often relies on ratings to evaluate the performance of workers as well as to qualify them for higher-paid work thus employment decisions may inherit consumers' biases [38]. Do algorithmic decisions such as search, ranking and matching determined by users' ratings also run the risk of reproducing human bias (see, e.g. [31, 32, 14] for related work in this direction)? What role does the design of online rating systems play in amplifying the effect of social bias? How does the amount and form of information provided by the platform about a worker affect hiring decisions? Furthermore, in a non-balanced market with more workers than employers, DM policy would also look different; intuitively, workers of lower probability to be high-skilled would stay unmatched for long periods of time. If employers were more than workers, $D$-identified employers could stay remain unmatched by the platform. In unbalanced markets, how does the effect of social bias change? And most generally, is there any platform policy that permanently eliminates discrimination?

### Acknowledgments

The authors would like to thank Daniela Saban and Ramesh Johari for helpful comments and discussions.

## Footnotes

[1]We do not make any assumption on the size of each social group. Alternatively, one may use the terms *privileged* and *unprivileged*.

[2] We refer to each worker as *he* and each employer as *she*.

[3] The *ex ante* idiosyncratic term $A_k$ is realized when employer $k$ meets the worker; the *ex post* term $P_k$ is realized after the employer hires the worker. Both $A_k$ and $P_k$ are independent of $Q$.

[4] This is the standard utility model in the related social learning literature (see e.g. [29, 9, 2])

[5]See Appendix A.1 for a formal description of systems dynamics.

[6]In comparison to the models considered in [9] and [41], there are also several technical differences in the utility function and the review structure.

[7]For a related result about a Bayesian social learning model, see Theorem 1 in [2].

[8]Note that $\Omega_k$ can be of any length 0 and $K - 1$.

[9]However, the optimality of the DM policy is an open, challenging question.

[10]Banning employers can possibly harm workers. Moreover, with more heterogeneity, other employers may seem to appear discriminating.

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
