[Supplementary Material · NIPS_discrimination-17.pdf]

# *Supplementary Material*:
# Discrimination in online markets: Effects of social bias on learning from reviews and policy design

**Faidra Monachou**
Stanford University
monachou@stanford.edu

**Itai Ashlagi**
Stanford University
iashlagi@stanford.edu

## Appendix

In Appendix A, we provide several definitions and details about the underlying large market continuum model that we consider in the main model in Section 2. In Appendices B and C, we include the proofs of all results in Sections 3 and 4, respectively. Appendix D contains auxiliary technical results used in the proofs. Finally, Appendix E discusses alternative models and extensions.

## A   Large Markets

### A.1   Large market setting

Time $t = 0, 1, \ldots$ is discrete but we assume that each arriving worker and employer stays for $K$ periods in the market. The market is initially empty and no workers and employers have entered the market before time $t = 0$.

**Worker and employer history.** At any time $t$, a mass $\lambda_i$ of agents $i \in \{A, B, N, D\}$ arrives in the market. For workers of each social group $c \in \{A, B\}$, a mass $\lambda_c^H = \lambda_c p_0$ consists of high-skilled workers and the rest $\lambda_c^L = \lambda_c(1 - p_0)$ of low-skilled workers.[1] Let $\Omega_k$, $k = 1, \ldots, K$ denote the worker history of length $k - 1$. Formally, the *worker history* $\Omega_k = \{\omega_1, \ldots, \omega_{k-1}\}$ consists of all the past hiring decisions $m_k \in \{0, 1\}$ and reviews $r_k \in \{\diamond, g, b\}$ for that worker, that is $\omega_k = (m_k, r_k)$. Let $H_n = \{h_1, \ldots, h_{n-1}\}$, $n = 1, \ldots, K$, also denote the *employer history* of length $n - 1$. Each $h_n$ consists of the hiring decision $m_n \in \{0, 1\}$ made by that employer about a worker with history $\Omega_k$[2] and social group $c \in \{A, B\}$, i.e. where $h_n = (m_n, \Omega_k, c)$. Initially, $H_1 = \emptyset$ and $\Omega_1 = \emptyset$.

**System profile.** We denote by $\mu(\Omega_k, c, Q)$, $k = 1, \ldots, K$, the mass of workers in the system with history $\Omega_k$, social group $c$ and skill level $Q$. Let $\nu(H_n, e)$ also be the mass of employers in the system with history $H_n$ belonging to group $e$. The evolution of the system is described by $(\mu_t, \nu_t)$ for $t \in \mathbb{N}$. A *matching policy* $\sigma(H_n, \Omega_k, c \mid (\mu_t, \nu_t))$ specifies the mass of workers of social group $c$ and history $\Omega_k$ to be matched to a mass of employers of history $H_n$, given the system state $(\mu_t, \nu_t)$ at time $t$.

At each period, employers meet workers according to the matching policy $\sigma$ (for example, in Section 3, we assumed a *uniform matching (UM) policy*). A mass of employers of group $e$ meets an equal mass of workers of social group $c$, true skill level $Q$ and history $\Omega_k$. Then, a fraction $\xi(\omega, \Omega_k, c, Q, e)$ of this mass leads to a *hiring-review outcome* $\omega \in \{(0, \diamond), (1, \diamond), (1, g), (1, b)\}$ (for example, see

(A.1)). The quantity $\xi(\omega, \Omega_k, c, Q, e)$ may be analytically computed based on any of the learning mechanisms (taste- or belief-based) described in Sections 2 and E.1.

For workers, all $\Omega_k$, $c$, $Q$ and $e$ determine the hiring review-outcome $\omega$. However, note that, given any of those mechanisms with constant prior belief, we know that, when an employer meets a candidate worker, only the history $\Omega_k$ and social group $c$ (and not skill level $Q$) of the worker play a role in her hiring decision; the true skill level $Q$ determines the review. For that reason, it is useful to compute the hiring probability $\pi(1, \Omega_k, c, e)$ separately; similarly, we denote the probability of not hiring by $\pi(0, \Omega_k, c, e)$. The distinction will become clear in equations (A.1) and (A.2).

## A.2 Dynamics, steady-state equilibrium and the UM policy

At any time $t$, a total mass $(\lambda_A + \lambda_B)K$ of workers and a total mass $(\lambda_N + \lambda_D)K$ of employers are present in the market. The platform cannot directly observe $\mu_t(\Omega_k, c, Q)$ and $\nu_t(H_n, e)$ but only $\sum_Q \mu_t(\Omega_n, c, Q)$ and $\sum_e \nu_t(H_n, e)$ for each history $\Omega_k$ and $H_n$. However, given $\lambda_i$, $\xi$ and $\pi$, the platform can *infer* each mass $\mu_t(\Omega_k, c, Q)$ and $\nu_t(H_n, g)$. Regarding workers, we have for $k = 1, \ldots, K - 1$, and absolute time $t + 1$ in the market,

$$\mu_{t+1}(\emptyset, c, Q) = \lambda_c^Q$$
$$\mu_{t+1}((\Omega_k, \omega), c, Q) = \mu_t(\Omega_k, c, Q) \sum_e \frac{\lambda_e}{\lambda_N + \lambda_D} \xi(\omega, \Omega_k, c, Q, e) \tag{A.1}$$

where the factor $\sum_e \frac{\lambda_e}{\lambda_N + \lambda_D} \xi(\omega, \Omega_k, c, Q, e)$ results from uniform matching.

Regarding employers, UM induces the following system dynamics for $n = 1, \ldots, K - 1$,

$$\nu_{t+1}(\emptyset, e) = \lambda_e$$
$$\nu_{t+1}((H_n, (m, \Omega_k, c)), e) = \nu_t(H_n, e) \sum_{\Omega_k, c, Q} \frac{\mu_t(\Omega_k, c, Q)}{(\lambda_A + \lambda_B)K} \pi(m, \Omega_k, c, e). \tag{A.2}$$

**Steady state of UM policy.** Under the above dynamics (A.1)-(A.2), the system yields a unique *steady-state equilibrium* $(\mu, \nu)$. The equilibrium $(\mu_{\text{UM}}, \nu_{\text{UM}})$ can be recursively computed for $n, k = 1, \ldots, K - 1$ from the following equations:

$$\mu_{\text{UM}}(\emptyset, c, Q) = \lambda_c^Q$$
$$\mu_{\text{UM}}((\Omega_k, \omega), c, Q) = \mu_{\text{UM}}(\Omega_k, c, Q) \sum_e \frac{\lambda_e}{\lambda_N + \lambda_D} \xi(\omega, \Omega_k, c, Q, e) \tag{A.3}$$

$$\nu_{\text{UM}}(\emptyset, e) = \lambda_e$$
$$\nu_{\text{UM}}((H_n, (m, \Omega_k, c)), e) = \nu_{\text{UM}}(H_n, e) \sum_{\Omega_k, c, Q} \frac{\mu_{\text{UM}}(\Omega_k, c, Q)}{(\lambda_A + \lambda_B)K} \pi(m, \Omega_k, c, e). \tag{A.4}$$

**Lemma A.1.** *The discrete time dynamical system* (A.1)-(A.2) *reaches a unique steady-state equilibrium.*

*Proof.* For simplicity, fix $c$ and $Q$. At first we focus only on $\mu_t$. For given $c$ and $Q$, the iteration mapping that describes the dynamics of $\mu_t$ is of the form $F(x) = Cx + d$. Each element $(\Omega_k, \omega)$, $\Omega_k$, $k = 1, \ldots, K - 1$ of the matrix $C$ is given by $C_{(\Omega_k, \omega), \Omega_k} = \sum_e \frac{\lambda_e}{\lambda_N + \lambda_D} \xi(\omega, \Omega_k, c, Q, e)$ while $C$ is zero everywhere else. For the vector $c$ we have that $d(\emptyset) = \lambda_c^Q$; $d$ takes a zero value everywhere else.

We have that

$$\|C\|_\infty = \max_{(\Omega, \omega)} C_{(\Omega, \omega), \Omega} = \sum_e \frac{\lambda_e}{\lambda_N + \lambda_D} \xi(\omega, \Omega, c, Q, e) < \sum_e \frac{\lambda_e}{\lambda_N + \lambda_D} = 1$$

since in each row all elements are zero expect for the element $((\Omega, \omega), \Omega)$ and, by Assumption 1, it follows that $0 < \xi(\omega, \Omega, c, Q, e) < 1$ for all $\omega \in \{(0, \diamond), (1, \diamond), (1, g), (1, b)\}$. Therefore, the spectral radius is also bounded by 1, i.e. $\rho(C) \leq \|C\|_\infty < 1$. Because any iteration of the form $\mu_{t+1} = C\mu_t + c$ converges for every starting point if and only if $\rho(C) < 1$, this system must converge. Furthermore, the limit has to be the fixed point of $F$.

We proved that $\mu_t$ converges to the steady-state equilibrium $\mu_{\text{UM}}$, and given the form of the dynamics (A.1), its convergence is independent of the convergence of $\nu_t$. We have that

$$\nu_{t+1} - \nu_{\text{UM}} < \frac{(\nu_t - \nu_{\text{UM}}) \cdot |\sum \mu_{\text{UM}}|}{(\lambda_A + \lambda_B)K} + \frac{\nu_t \cdot |\sum \mu_t - \sum \mu_{\text{UM}}|}{(\lambda_A + \lambda_B)K}$$

Since $\mu_t \to \mu_{\text{UM}}$ as $t \to \infty$ (note that $\nu_t$ is bounded for any fixed $K$) and $|\sum \mu_{\text{UM}}| = (\lambda_A + \lambda_B)K$, we can eventually show that

$$\|\nu_{t+1} - \nu_{\text{UM}}\| \to 0.$$

Therefore, we get that $\nu_t$ also converges to the steady-state equilibrium $\nu_{\text{UM}}$. ☐

# B Proofs from Section 3

***Proof of Lemma 1.*** At period $k$, we have

$$q_k^N - q_k^D = \frac{(1 - \beta)G_0}{B_k + G_k + N_0} > 0. \tag{B.1}$$

Therefore, as $k$ increases, $B_k + G_k$ also weakly increases thus $q_k^N - q_k^D$ is weakly decreasing. ☐

***Proof of Lemma 2.*** It follows directly from Lemma 1 and the fact that the CDF $F_A$ is strictly increasing on the support $[\underline{a}, \overline{a}]$. ☐

***Proof Sketch of Theorem 1.*** The existence of the unique steady-state equilibrium of the market is proved in Lemma A.1.

Given this result, we compare the expected welfare of a minority and a majority worker with same skill level $Q$ over their lifetime to show the existence of discrimination. For clarity of exposition, we assume that the probability of receiving a review conditional on being hired is $\eta = 1$, as it is straightforward to generalize the proof for general $\eta > 0$. As we define in Appendix A.1, the worker history $\Omega_k = \{\omega_1, \ldots, \omega_{k-1}\}$ consists of all the past hiring decisions $m_k \in \{0, 1\}$ and reviews $r_k \in \{\diamond, g, b\}$ for that worker, that is $\omega_k = (m_k, r_k)$. Initially, $\Omega_1 = \emptyset$.

We first analyze the expected welfare for the fixed $Q \in \{H, L\}$, and use backward induction on the history length $k$ to show that, given a worker history $\Omega_k$, the expected future welfare of majority workers after life period $k$ and up to the last period $K$ is larger than the one of majority workers. The basis step (period $K$) is trivial and follows directly by Lemma 2. Then, by a slight abuse of notation and given history $\Omega_{k-1}$, we can write the expected welfare of a worker of group $c$ after life period $k - 1$ in the following form

$$W_Q^c(\Omega_{k-1}, K) = \mathbb{P}(\omega_k = (1, g) \mid Q, c, (\Omega_{k-1}, (1, g))) W_Q^c((\Omega_{k-1}, (1, g)), K) +$$
$$\mathbb{P}(\omega_k = (1, b) \mid Q, c, (\Omega_{k-1}, (1, b))) W_Q^c((\Omega_{k-1}, (1, b)), K) +$$
$$(1 - \mathbb{P}(\omega_k = (1, g) \mid Q, c, (\Omega_{k-1}, (1, g))) - \mathbb{P}(\omega_k = (1, b) \mid Q, c, (\Omega_{k-1}, (1, b)))) W_Q^c((\Omega_{k-1}, (0, \diamond)), K)$$

where the last term can be written as

$$W_Q^c((\Omega_{k-1}, (0, \diamond)), K) = \delta W_Q^c(\Omega_{k-1}, K) - \psi(c, Q, \Omega_{k-1}),$$

where the term $\psi(c, Q, \Omega_{k-1})$ refers to the last period $K$ only.

By the induction hypothesis (for $k$), we have that

$$W_Q^A((\Omega_{k-1}, (1, g)), K) > W_Q^B((\Omega_{k-1}, (1, g)), K),$$

$$W_Q^A((\Omega_{k-1}, (1, b)), K) > W_Q^B((\Omega_{k-1}, (1, b)), K),$$

$$W_Q^A((\Omega_{k-1}, (0, \diamond)), K) > W_Q^B((\Omega_{k-1}, (0, \diamond)), K).$$

By Lemma D.4,

$$\mathbb{P}(\omega_k = (1, g) \mid A, c, (\Omega_{k-1}, (1, g))) \geq \mathbb{P}(\omega_k = (1, g) \mid B, c, (\Omega_{k-1}, (1, g)))$$

and

$$\mathbb{P}(\omega_k = (1, b) \mid A, c, (\Omega_{k-1}, (1, b))) \geq \mathbb{P}(\omega_k = (1, b) \mid B, c, (\Omega_{k-1}, (1, b))),$$

which hold with strict inequality for some (if not all) histories $\Omega_{k-1}$. Assuming $\delta$ small enough and using the previous inequalities, it follows easily that $W_Q^A(\Omega_{k-1}, K) > W_Q^B(\Omega_{k-1}, K)$.

Now, it remains to generalize the result for any $\delta \in (0, 1)$ and apply it for $W_Q^c(K) = W_Q^c(\emptyset, K)$. Fix $K$. We adopt the following construction argument. We start with all histories $\Omega_{K-1}$ and feasible $\delta'$, take inequality $W_Q^A(\Omega_{K-1}, K) > W_Q^B(\Omega_{K-1}, K)$ and multiply by $(\delta/\delta')^K$. Then, we repeat the same process for lifetime $K - 1$ and all histories $\Omega_{K-1}$, and continue up to lifetime 1 and empty history $\Omega_1 = \emptyset$. Adding all the formed inequalities and starting from the empty history $\Omega_1 = \emptyset$, we ended up with $W_Q^A(K) > W_Q^B(K)$ which concludes the proof. $\qquad\square$

***Proof of Theorem 2.*** We consider the case of minority and majority workers separately.

At any time $k$ in a worker's life, both groups of employers have the same belief $q_k$ about the same majority worker. By (9), it suffices to look only at the sequence $q_{\tau_1+1}, q_{\tau_2+1}, \ldots$. For convenience, we define $\hat{q}_{\tau_k}^e \triangleq q_{\tau_k+1}^e$. An application of Lemma D.1 (coupled with Assumption 1) guarantees that $m_t = 1$ infinitely often. Therefore, to show that $q_k \to q_\infty$ almost surely, it suffices to prove that $\hat{q}_{\tau_k} \to q_\infty$ almost surely as $k$ grows to $\infty$.

We use a "Robbins-Monro" argument (adapted from Theorem 2 in [6] to our model). For $k \geq 1$, we can write that

$$\hat{q}_{\tau_{k+1}} = \hat{q}_{\tau_k} + \frac{1}{k + 1 + N_0}(\mathbf{1}_{\{r_{\tau_{k+1}}=g\}} - \hat{q}_{\tau_k}). \tag{B.2}$$

Equation (B.2) describes a stochastic approximation algorithm of the form (D.2) where

$$\begin{aligned} h(q) &= \mathbb{E}(\mathbf{1}_{\{r_{\tau_{k+1}}=g\}} - \hat{q}_{\tau_k} \mid \hat{q}_{\tau_k} = q, Q) \\ &= \mathbb{P}(r_{\tau_{k+1}} = g \mid \hat{q}_{\tau_k} = q, Q) - q, \end{aligned} \tag{B.3}$$

and the sequence of step gains is $\gamma_{\tau_k+1} = \frac{1}{k+1+N_0}, k \geq 1$.

Next we show that $h(q)$ satisfies the assumptions in Lemma D.2. Specifically,

$$|H(q, X_{\tau_{k+1}})| = |\mathbf{1}_{\{r_{\tau_{k+1}}=g\}} - q| \leq \max\{|q|, |1 - q|\}$$

implies that condition (D.4) holds, i.e.

$$\sigma^2(q) \leq C(1 + |q|^2)$$

for some constant $C > 0$.

To prove the stability condition (D.5), first observe that the function $h(q)$ is strictly decreasing in $q$. Furthermore, $\lim_{q \to \infty} h(q) = -\infty$ while Assumption 1 guarantees that $h(0) > 0$. This ensures that $h(q) = 0$ has a unique solution $q_*$. Hence, by the fact that $h(q)$ strictly decreasing in $q$, we can easily show that,

$$(q - q_*)h(q) = (q - q_*)(h(q) - h(q_*)) \leq -(q - q_*)^2.$$

This in turn implies that for all $\epsilon > 0$,

$$\sup_{\epsilon \leq |q-q_*| \leq \frac{1}{\epsilon}} (q - q_*)h(q) \leq \sup_{\epsilon \leq |q-q_*| \leq \frac{1}{\epsilon}} -(q - q_*)^2 \leq -\epsilon^2 < 0.$$

All conditions of Lemma D.2 are satisfied thus we conclude that $\hat{q}_{\tau_k}^e$ (and consequently $q_k^e$) converges almost surely to the limit $q_\infty^e = q_*$, i.e. for both $g \in \{N, D\}$, $q_k^e \to q_*$ almost surely.

For the second part of the theorem, we have proved that $q_*$ satisfies $h(q_*) = 0$. Hence, from (B.3) we find $q_\infty(Q) \equiv q_*$ for $Q \in \{H, L\}$. Also, it is clear that $h(q)$ and $q_\infty(Q)$ do not depend on the prior belief of each employer group.

We also need to prove the theorem for the case of minority workers which is more complicated.

Specifically, by Lemma 1, we have that $\hat{q}^N_{\tau_k} > \hat{q}^D_{\tau_k}$, $k \geq 1$. However, for each employer group $e \in \{N, D\}$, an analog of equation (B.2) holds for each $e \in \{N, D\}$ and for all $k \geq 1$,

$$\hat{q}^e_{\tau_{k+1}} = \hat{q}^e_{\tau_k} + \frac{1}{k+1+N_0}(\mathbf{1}_{\{r_{\tau_{k+1}}=g\}} - \hat{q}^e_{\tau_k}) \tag{B.4}$$

but here the probability

$$\mathbb{P}(r_{\tau_{k+1}} = g \mid \hat{q}^N_{\tau_k} = q^N, \hat{q}^D_{\tau_k} = q^D, Q) = \tag{B.5}$$

$$\frac{\lambda_N}{\lambda_N + \lambda_D}\mathbb{P}(r_{\tau_{k+1}} = g \mid \hat{q}^N_{\tau_k} = q^N, Q) + \frac{\lambda_D}{\lambda_N + \lambda_D}\mathbb{P}(r_{\tau_{k+1}} = g \mid \hat{q}^D_{\tau_k} = q^D, Q)$$

depends on both $\hat{q}^N_{\tau_k} = q^N$ and $\hat{q}^D_{\tau_k} = q^D$.

Lemma D.2 can now be applied to the pair of random variables $\hat{q}^N_{\tau_k}$ and $\hat{q}^D_{\tau_k}$. The rest of the proof follows similar steps as in the case of majority workers and leads to the same conclusion, i.e. that for both $e \in \{N, D\}$, $\hat{q}^N_{\tau_k} \to q_\infty(Q)$ almost surely. Intuitively, as $k$ grows, Lemma 1 implies that the difference $\hat{q}^N_{\tau_k} - \hat{q}^D_{\tau_k}$ tends to 0, which means that algorithm (B.4) resembles the behavior of (B.2).

For the last statement of the theorem, by a contradiction argument, we can show that

$$q_\infty(H) > q_\infty(L).$$

$\square$

***Proof Sketch of Proposition 3.*** Since $\bar{a} + \mu_P < 0$, then for $\hat{q}$ small enough, we have

$$\mathbb{P}(A_k + \mu_P + \hat{q} \leq 0) = 1. \tag{B.6}$$

However, in finite time $t_0$, there is a positive probability that $q^N_t$ can become as small as $\hat{q}$. Then, (B.6) implies that hirings stop. By (9), we conclude that $q^e_k$ remains unchanged, i.e. $q^e_k = q^e_{k_0+1} \leq \hat{q}$, for all $k > k_0$.
$\square$

## C  Proofs from Section 4

**Lemma C.1** (Lemma 4). *Suppose that the mass $\mu(\Omega_k, B, Q)$, $k = 1, \ldots, K'$ of minority workers in the system is known for each history $\Omega_k$ and skill level $Q$, and does not change over time. A fixed employer is paired uniformly at random to a minority worker $n$, i.i.d. for each period $n$ of the employer's lifetime. Then, for large enough $K$ and $\theta$, the expected time until an employer of group $e \in \{N, D\}$ gets e-labelled is at most $K$, i.e.*

$$\mathbb{E}(\inf\{n > 0 : l_n \geq \theta\} \mid e = D) < K \text{ and } \mathbb{E}(\inf\{n > 0 : l_n \leq -\theta\} \mid e = N) < K. \tag{C.1}$$

*Proof.* First, we analyze further the learning problem of the platform regarding the true type $e \in \{N, D\}$ of our fixed employer. When a minority worker of history $\Omega_k$ is matched to the employer then the log-likelihood ratio $l_n$ of learning the type of the employer is updated as follows:

$$l_{n+1} = l_n + \log \frac{X^D(m_n, \Omega_k)}{X^N(m_n, \Omega_k)}, n \geq 1 \tag{C.2}$$

where $l_1 = \log \frac{\lambda_D}{\lambda_N}$ and

$$X^e(m_n, \Omega_k) \triangleq m_n \pi(1, \Omega_k, B, e) + (1 - m_n)\pi(0, \Omega_n, B, e).$$

The probabilities $\pi(1, \Omega_k, B, e)$ and $\pi(0, \Omega_k, B, e)$ have been defined in Appendix A.1 and represent the probability that an employer of group $e$ hires or not a minority worker with history $\Omega_k$, respectively.

Next, we adapt Lemma D.5 to our setting. Specifically, let $I \equiv \{N, D\}$, $X_n \equiv (m_n, \Omega_k)$,

$$f^{(N)}(X_n) \equiv \log \frac{X^D(m_n, \Omega_k)}{X^N(m_n, \Omega_k)}, f^{(D)}(X_n) \equiv \log \frac{X^N(m_n, \Omega_k)}{X^D(m_n, \Omega_k)}.$$

By Lemma 2 and Assumption 1, the KL-divergence[3]

$$\text{KL}(D, N \mid \Omega_k) \triangleq \pi(1, \Omega_k, B, D) \log\left(\frac{\pi(1, \Omega_k, B, D)}{\pi(1, \Omega_k, B, N)}\right) + \pi(0, \Omega_k, B, D) \log\left(\frac{\pi(0, \Omega_k, B, D)}{\pi(0, \Omega_k, B, N)}\right)$$

is positive for some (if not all) $\Omega_k$, $n = 1, \ldots, N$. (Specifically, $\text{KL}(D, N \mid \Omega_k)$ is positive for some (if not all) $\Omega_k$ for each history length $k$). The same holds for $\text{KL}(N, D \mid \Omega_k)$ which is defined in a symmetric way. Also note that minority workers are chosen uniformly at random so any worker of any history $\Omega_k$, $n = k, \ldots, K'$ has positive probability to be matched to that particular employer. Thus, given $e = D$,

$$\mathbb{E}(f^{(N)}(X_n)) = \sum_{\Omega_k} \frac{\sum_Q \mu(\Omega_k, B, Q)}{\sum_{\Omega_k, Q} \mu(\Omega_k, B, Q)} \text{KL}(D, N \mid \Omega_k)$$

while the formula for $\mathbb{E}(f^{(D)}(X_n))$ follows similarly. By applying Lemma D.5 for $\alpha \equiv \theta$, we get that there exists a large enough $\theta$ such that

$$\mathbb{E}(\inf\{n > 0 : l_n \geq \theta\} \mid e = D) < A \log \theta$$

for some constant $A$ that does not depend on $\theta$. Choosing a large enough $K > A \log \theta$ completes the proof. □

**Lemma C.2.** *Under the DM policy, there exists a steady-state equilibrium in the market. Furthermore, for large enough $\theta_N$ and $\theta_D$, there exists a steady-state equilibrium where the steady-state fraction of $D$ employers in the learning pool is strictly smaller than $\frac{\lambda_D}{\lambda_N + \lambda_D}$.*

*Proof sketch.* We use notation and definitions provided in Appendix A.1.

We write down the dynamics of the system under the DM policy. In the technical details of the proof, we can ignore majority workers and employers matched to them; establishing that an equilibrium point exists for minority workers' mass profile directly generalizes the result to the whole system. For convenience, we also omit $B$ from notation.

Let $\mathcal{H}_e^+$ denote the set of all histories such that employers of group $e$ are correctly identified as $e$. Similarly, let $\mathcal{H}_e^-$ denote the set of all histories such that employers of group $e$ are incorrectly identified as $e' \neq e$. For minority workers in the learning pool (LP), we have that for $k = 1, \ldots, K$:

$$\mu_{t+1}^{\text{LP}}(\emptyset, Q) = \mu_t^{\text{LP}}(\emptyset, Q) \min\left\{1, \left[1 - \frac{\sum_n \nu_t^{\text{ID}}(n) + \sum_{H_n \in \mathcal{H}_N^+ \cup \mathcal{H}_D^-} \nu_t^{\text{LP}}(H_n, N) - \sum_k \mu_t^{\text{ID}}(k, Q)}{\sum_{\Omega, Q} \mu_t^{\text{LP}}(\Omega, Q)}\right]^+\right\}$$

(C.3)

$$\mu_{t+1}^{\text{LP}}((\Omega_k, \omega), Q) = \mu_t^{\text{LP}}(\Omega_k, Q) \sum_e \frac{\sum_{H_n} \nu_t^{\text{LP}}(H_n, e)}{\sum_e \sum_{H_n} \nu_t^{\text{LP}}(H_n, e)} \xi(\omega, \Omega_k, Q, B, e) \min\left\{1, \left[1 - \frac{\sum_n \nu_t^{\text{ID}}(n) + \sum_{H_n \in \mathcal{H}_N^+ \cup \mathcal{H}_D^-} \nu_t^{\text{LP}}(H_n, N) - \sum_k \mu_t^{\text{ID}}(k, Q)}{\sum_{\Omega, Q} \mu_t^{\text{LP}}(\Omega, Q)}\right]^+\right\}$$

(C.4)

where by (ID) we denote the mass of $N$-identified employers and the mass of minority workers matching to each other out of the learning pool.

For employers in the learning pool, we have for $n = 1, \ldots, K$:

$$\nu_{t+1}^{\text{LP}}(\emptyset, e) = \frac{\lambda_e}{\lambda_N + \lambda_D}\left(\sum_{H_{K+1}} \sum_{e'} \nu_t^{\text{LP}}(H_{K+1}, e') + \sum_{H_n \in \mathcal{H}_N^+} \nu_t^{\text{LP}}(H_n, N) + \sum_{H_n \in \mathcal{H}_D^-} \nu_t^{\text{LP}}(H_n, D)\right.$$
$$\left. - \sum_{\Omega_k} \mu_t^{\text{LP}}(\Omega_k, Q)(1 - \min\{1, \left[1 - \frac{\sum_n \nu_t^{\text{ID}}(n) + \sum_{H_n \in \mathcal{H}_N^+ \cup \mathcal{H}_D^-} \nu_t^{\text{LP}}(H_n, N) - \sum_k \mu_t^{\text{ID}}(k, Q)}{\sum_{\Omega, Q} \mu_t^{\text{LP}}(\Omega, Q)}\right]^+\})\right)$$

(C.5)

$$\nu_{t+1}^{\text{LP}}((H_n,(m,\Omega_k)),e) = \nu_t^{\text{LP}}(H_n,e)\sum_Q \frac{\mu_t^{\text{LP}}(\Omega_k,Q)}{\sum_{\Omega_k,Q}\mu_t^{\text{LP}}(\Omega_k,Q)}\pi(m,\Omega_k,B,e), \text{ if } H_n \notin \mathcal{H}_e^+ \cup \mathcal{H}_e^-.$$
(C.6)

For $N$-identified employers and minority workers matching to each other out of the learning pool (ID), we care only about their history lengths (in order to establish the proof). By slightly abusing the notation, we can group their mass as follows

$$\mu_{t+1}^{\text{ID}}(0,Q) = \lambda_B - \sum_{\Omega_k}\sum_Q \mu_t^{\text{LP}}(\emptyset,Q)$$
(C.7)

$$\mu_{t+1}^{\text{ID}}(k+1,Q) = \mu_t^{\text{ID}}(k,Q) + \sum_{\Omega_k}\mu_t^{\text{LP}}(\Omega_k,Q)(1 - \min\{1, \left[1 - \frac{\sum_n \nu_t^{\text{ID}}(n) + \sum_{H_n\in\mathcal{H}_N^+\cup\mathcal{H}_D^-}\nu_t^{\text{LP}}(H_n,N) - \sum_k\mu_t^{\text{ID}}(k,Q)}{\sum_{\Omega,Q}\mu_t^{\text{LP}}(\Omega,Q)}\right]^+\})$$
(C.8)

$$\nu_{t+1}^{\text{ID}}(n+1) = \nu_t^{\text{ID}}(n) + \sum_{H_n\in\mathcal{H}_N^+}\nu_t^{\text{LP}}(H_n,N) + \sum_{H_n\in\mathcal{H}_D^-}\nu_t^{\text{LP}}(H_n,D)$$
(C.9)

for $n,k = 1,\ldots,K$.

From equation (C.3), it becomes clear that at any equilibrium point $(\mu_{\text{LP}},\nu_{\text{LP}},\mu_{\text{ID}},\nu_{\text{ID}})$ we must have

$$\min\{1, \frac{\sum_k \mu_t^{\text{ID}}(k,Q)}{\sum_n \nu_t^{\text{ID}}(n) + \sum_{H_n\in\mathcal{H}_N^+}\nu_t^{\text{LP}}(H_n,N) + \sum_{H_n\in\mathcal{H}_D^-}\nu_t^{\text{LP}}(H_n,D)}\} = 1$$

thus all the system equations are simplified. By (C.8) and (C.9), it also follows that the total sum of minority workers and the total sum of employers out of the learning pool (ID) remains constant. The same follows for minority workers and employers in the learning pool; furthermore, their total mass is equal, i.e.

$$K\sum_Q \mu_{\text{LP}}(\emptyset,Q) = \sum_{H_n,e}\nu_{\text{LP}}(H_n,e).$$
(C.10)

To compute the fixed point $(\mu_{\text{LP}},\nu_{\text{LP}},\mu_{\text{ID}},\nu_{\text{ID}})$ we proceed as follows. We start with equation (C.4) and observe that the value of $\mu_{\text{LP}}$ depends on the fraction of $D$ employers in the learning pool, i.e. the factor $j(D) \triangleq \frac{\sum_{H_n}\nu_{\text{LP}}(H_n,D)}{\sum_e\sum_{H_n}\nu_{\text{LP}}(H_n,e)}$, which is constant at the equilibrium point.

Our goal for the welfare of minority workers in the learning pool is to improve it or at least keep it the same as in UM. Suppose that we choose a target $j(D) \leq \frac{\lambda_D}{\lambda_D + \lambda_N}$. Solving for $\mu_{\text{LP}}$, we can compute $\mu_{\text{LP}}$ as a function of our chosen $j(D)$ and quantities $\mu_{\text{LP}}(\emptyset,H),\mu_{\text{LP}}(\emptyset,L)$. Note that arguments similar to the ones in the proof of Lemma A.1 prove that, given any $0 \leq j(D) \leq 1$, there exists a unique solution $\mu_{\text{LP}}$ to (C.3)-(C.4). We can compute this solution by constructively starting from $\Omega_1 = \emptyset$. Substituting each factor $\frac{\mu_{\text{LP}}(\Omega_k,Q)}{K\sum_Q \mu_{\text{LP}}(\emptyset,Q)}$ in (C.6), we also find a unique solution $\nu_{\text{LP}}$.

Forming the equation

$$j(D) = \frac{\sum_{H_n}\nu_{\text{LP}}(H_n,D)}{K\sum_Q \mu_{\text{UM}}(\emptyset,Q)},$$
(C.11)

we find that the RHS in (C.11) is linear in $j(D)$ thus it has at most one solution. If it has a solution $j(D) \leq \frac{\lambda_D}{\lambda_N}$, then we have found a desired equilibrium point. If not, then, for the second part of the lemma statement, we still need to prove that, we can always find large enough $K, \theta_D$ and $\theta_N$, such that the system (C.3)-(C.9) has a fixed point solution with $j(D) \leq \frac{\lambda_D}{\lambda_N + \lambda_D}$ leading to an equilibrium point that strictly improves the welfare of minority workers.

We construct such a solution as follows. The most important step is to define a high enough $\theta_N$ such that, given our chosen $K$, no employer $D$ or $N$ can get identified as $N$. Since $K$ is finite, this is always feasible for large enough $\theta_N$. Next, we need to specify a large enough $\theta_D$, so that the fraction of $D$ employers who get identified as $D$ is strictly larger than $\frac{\lambda_D}{\lambda_N + \lambda_D}$. By Lemma D.4, we can find

such a $\theta_D$. The intuition is as follows. Consider only employers who have rejected all the workers they were matched to. Then, by Lemma D.4 and the fact that all employers in the learning pool have the same probability to get matched to workers with histories $\Omega^1, \ldots, \Omega^n$, we get that

$$\mathbb{P}(H_n = \{(0, \Omega^1, B), \ldots, (0, \Omega^{n-1}, B)\} \mid e = D) > \mathbb{P}(H_n = \{(0, \Omega^1, B), \ldots, (0, \Omega^{n-1}, B)\} \mid e = N) \tag{C.12}$$

with strict inequality for at least some $\Omega^1, \ldots, \Omega^{n-1}$ (for example, $\Omega^1 = \ldots = \Omega^{n-1} = \emptyset$).

Then, consider the set $\mathcal{H}_D$ of all employer histories $H_n$ such that an employer gets $D$-identified[4], i.e.

$$\mathcal{H}_D = \{H_n = \{h_1, \ldots, h_{n-1}\}, n \in \mathbb{N} : l_n(H_n) > \theta_D \text{ and } l_{n'}(H'_n) \leq \theta_D, \forall H_{n'} = \{h_1, \ldots, h_{n'-1}\}, n' < n\}. \tag{C.13}$$

For $\theta_D$ large enough, the set $\mathcal{H}_D$ contains only a subset of the histories of the form $H_n = \{(0, \Omega^1, \diamond), \ldots, (0, \Omega^{n-1}, \diamond)\}$. Therefore, by equation (C.12), it follows that

$$\sum_{H_n \in \mathcal{H}_D} \nu_{\mathrm{LP}}(H_n, D) = \nu_{\mathrm{LP}}(\emptyset, D) \sum_{H_n \in \mathcal{H}_D} \mathbb{P}(H_n \mid e = D) > \nu_{\mathrm{LP}}(\emptyset, D) \sum_{H_n \in \mathcal{H}_D} \mathbb{P}(H_n \mid e = N) = \frac{\lambda_D}{\lambda_N} \sum_{H_n \in \mathcal{H}_D} \nu_{\mathrm{LP}}(H_n, N).$$

Equivalently, we get the result we wanted, i.e $j(D) \leq \frac{\lambda_D}{\lambda_D + \lambda_N}$. $\qquad\square$

**Lemma C.3.** *Given a threshold $\theta > 0$, the probability that a $D$ employer is incorrectly $N$-identified is*

$$\mathbb{P}(l_n \leq -\theta \text{ for some } n \leq K \mid g = D) \leq \frac{\lambda_N / \lambda_D}{\theta} \tag{C.14}$$

*while the same probability for $N$ employers is also*

$$\mathbb{P}(l_n \geq \theta \text{ for some } n \leq K \mid g = N) \leq \frac{\lambda_N / \lambda_D}{\theta}. \tag{C.15}$$

*Proof.* It follows from the fact that the likelihood ratio $\frac{\mathbb{P}(g=D|H_n)}{\mathbb{P}(g=N|H_n)}$ is a Martingale and we can apply Doob's martingale inequality to prove it. $\qquad\square$

***Proof Sketch of Theorem 3.*** At the steady-state equilibrium of the market (see the proof of Lemma C.2 for details), a fraction $x_{\mathrm{ID}} \geq 0$ of the incoming minority workers is matched at least to some $N$-identified employers. A fraction $x_{\mathrm{UM}} \geq 0$ of the incoming minority workers is randomly matched $N$ or $D$ (unidentified) employers where he matches to $D$ employers with probability $\frac{\lambda_D}{\lambda_D + \lambda_N}$ (exactly as under UM). The rest $1 - x_{\mathrm{ID}} - x_{\mathrm{UM}} > 0$ enter the learning pool.

An incoming minority worker who will be matched with $N$-identified employers (with probability $x_{\mathrm{ID}}$) will have a higher welfare in expectation. More specifically, by Lemma C.3, a $D$ employer incorrectly becomes $N$-identified with probability

$$\mathbb{P}(l_n \leq -\theta_N \text{ for some } n \leq K \mid g = D) \leq \frac{\lambda_N / \lambda_D}{\theta_N},$$

meaning that at most a fraction $\frac{\lambda_N / \lambda_D}{\theta_N}$ of the $D$ employers are actually get identified as $N$. For large enough $\theta_D$ and $\theta_N$, the probability to match with a $D$ employer who was identified as $N$ is smaller than the probability $\frac{\lambda_D}{\lambda_D + \lambda_N}$ to match with a $D$ employer under the UM policy.

Similarly to the proof of Theorem 1, we can show that, given the fact that an incoming minority worker of skill level $Q$ has been selected to match mostly with $N$-identified employers, his expected welfare $W_{Q,\mathrm{ID}}^B$ increases compared to his welfare $W_{Q,\mathrm{UM}}^B$ under UM.

On the other hand, we can control (via $\theta_N$ and $\theta_D$, if necessary; see Lemma C.2 for details) the fraction $j(D)$ of $D$ employers who are in the learning pool at each time $t$ so that $j(D) \leq \frac{\lambda_D}{\lambda_D + \lambda_N}$. Thus, an incoming minority worker who enters the learning pool (with probability $1 - x_{\mathrm{ID}} - x_{\mathrm{UM}}$) will have better expected welfare $W_{Q,\mathrm{LP}}^B$ than the welfare $W_{Q,\mathrm{UM}}^B$ he had under the UM policy.

Finally, putting everything together, we get that

$$W_{Q,\mathrm{DM}}^B(K) = x_{\mathrm{ID}} W_{Q,\mathrm{ID}}^B + x_{\mathrm{UM}} W_{Q,\mathrm{UM}}^B + (1 - x_{\mathrm{ID}} - x_{\mathrm{UM}}) W_{Q,\mathrm{LP}}^B > W_{Q,\mathrm{UM}}^B.$$

Consequently, the discrimination gap decreases, i.e.

$$d_{\mathrm{DM}}(Q, K) < d_{\mathrm{UM}}(Q, K).$$

$\qquad\square$

# D Auxiliary Lemmas

**Lemma D.1** (Theorem 5.3.2 in [9]). *Let $\mathcal{F}_t$ be a filtration with $\mathcal{F}_0 = \{\emptyset, \Omega\}$ and $E_t$, $t \geq 1$ a sequence of events with $E_t \in \mathcal{F}_t$. Then,*

$$\{E_t \ i.o.\} = \{\sum_{t=1}^{\infty} \mathbb{P}(E_t \mid \mathcal{F}_{t-1}) = \infty\} \tag{D.1}$$

**Lemma D.2** (Theorem 1 in Appendix to Part II in [5]). *Let $\mathcal{F}_t$ denote the $\sigma$-field of events generated by the random variables $\theta_1, X_1, \dots, \theta_t, X_t$. Consider the* Robbins-Monro *stochastic approximation algorithm*

$$\theta_{t+1} = \theta_t + \gamma_{t+1} H(\theta_t, X_{t+1}) \tag{D.2}$$

*where*

$$\mathbb{E}(H(\theta_t, X_{t+1}) - h(\theta_t) \mid \mathcal{F}_t) = 0 \tag{D.3}$$

*with $h(\theta) = \int H(\theta, x)\mu_\theta(dx)$. Specifically, suppose that*

$$\sigma^2(\theta) = \int |H(\theta, x)|^2 \mu_\theta(dx) \leq C(1 + |\theta|^2) \tag{D.4}$$

*for some constant $C$, as well as that the following stability condition holds*

$$\exists \theta_* : \sup_{\epsilon \leq |\theta - \theta_*| \leq \frac{1}{\epsilon}} (\theta - \theta_*)^T h(\theta) < 0 \text{ for all } \epsilon > 0. \tag{D.5}$$

*With the above assumptions, if the sequence $\{\gamma_t\}_{t\geq 1}$ satisfies $\sum \gamma_t = \infty$, $\sum \gamma_t^2 < \infty$, then $\theta_t \to \theta^*$ almost surely.*

**Lemma D.3** (Theorem 22 in Section 1.1.10 in [5]). *Let $\gamma_t = \frac{A}{n^a + B}$ for some $0 \leq a \leq 1$. Under the assumptions in Lemma D.2, the algorithm* (D.2) *has the following property*

$$\mathbb{E}(|\theta_t - \theta_*|^2) \leq \kappa(a)\gamma_t \tag{D.6}$$

*for some suitable constant $\kappa(a)$.*

**Lemma D.4.** *Under the naive social learning rule* (2)*, the following properties hold:*

   (i) $\mathbb{P}((1, g) \mid Q, \Omega_k, g = N) \geq \mathbb{P}((1, g) \mid Q, \Omega_k, g = D)$

   (ii) $\mathbb{P}((1, b) \mid Q, \Omega_k, g = N) \geq \mathbb{P}((1, b) \mid Q, \Omega_k, g = D)$

   (ii) $\mathbb{P}((1, \diamond) \mid Q, \Omega_k, g = N) \geq \mathbb{P}((1, \diamond) \mid Q, \Omega_k, g = D)$

*(For at least $q_k^D \leq q_0$, we also have strict inequality.)*

*Proof.* (i) Observe that

$$\mathbb{P}((1, g) \mid Q, \Omega_k, g) = \mathbb{P}(A_k + \mathbf{1}_{\{Q=H\}} + P_k \geq 0, A_k + q_k^e + \mu_P \geq 0) =$$

$$= \int_{\max(\underline{a}, -\mu_P - q_k^e)}^{\overline{\alpha}} \int_{-a - \mathbf{1}_{\{Q=H\}}}^{\overline{p}} f_A(a) f_P(p) \, dp \, da.$$

Since $q_k^N > q_k^D$ by Lemma 1, we get that

$$\mathbb{P}((1, g) \mid Q, \Omega_k, g = N) - \mathbb{P}((1, g) \mid Q, \Omega_k, g = D) =$$

$$\int_{\max(\underline{a}, -\mu_P - q_k^N)}^{\max(\underline{a}, -\mu_P - q_k^D)} \int_{-a - \mathbf{1}_{\{Q=H\}}}^{\overline{p}} f_A(a) f_P(p) \, dp \, da \geq 0.$$

(ii), (iii) Similar to (i).

We note that, unless $\underline{a} \geq -\mu_P - q_k^D$, all parts hold with strict inequality. By Assumption 1, for $q_k^D \leq q_0$, we always have strict inequality.

$\square$

**Lemma D.5** (Lemma 4.3 in [3], Lemma A.3 in [11]). *Let $X_1, X_2, \ldots$ be i.i.d random variables on some finite state space $\mathcal{X}$ with marginals $p(x)$. Let $f^{(i)} : \mathcal{X} \to \mathbb{R}$ such that $0 < \mathbb{E}(f^{(i)}(X_i)) < \infty$, $i \in I$ where $I$ is finite. Let $S_n^{(i)} = f^{(i)}(X_1) + \ldots + f^{(i)}(X_n)$, $L_a^{(i)} = \sum_{n=1}^{\infty} \mathbf{1}\{\inf_{t \geq n} S_t^{(i)} \leq a\}$ and $L_a = \max_{i \in I} L_a^{(i)}$. Then,*

$$\limsup_{\alpha \to \infty} \frac{\mathbb{E}(L_a)}{\alpha} \leq \frac{1}{\min_{i \in I} \mathbb{E}(f^{(i)}(X_i))}. \tag{D.7}$$

# E    Extensions: Social bias and belief updating

## E.1    Taste-based social bias

In this section, we focus on the case of taste-based social bias. As we discuss next, discrimination persists even under this type of bias but learning which employer is discriminating is inherently easier.

In contrast to the case of belief-based discrimination, discriminating employers with taste-based bias have the same prior belief $q_0 = \frac{G_0}{N_0}$ about minority and majority workers but will *knowingly* discriminate against minority workers due to pure preference. Specifically, we consider that, if the employer $k$ of group $e$ hires the worker of social group $c$, she receives utility

$$U_k = A_k + \mathbf{1}_{\{Q=H\}} + P_k + r_c^e, \tag{E.1}$$

where $\rho_c^e$ is an employer group-specific *taste parameter* about race.[5] We normalize $r_B^N = r_A^N = r_A^D = 0$ and $r_B^D = -r < 0$. Furthermore, the ex ante idiosyncratic term $A_k$, ex post idiosyncratic term $P_k$ and constant $r$ should satisfy the richness assumption.[6]

It follows that, if social bias is taste-based, then inequality in hiring probabilities persists even under perfect information. On the one hand, discrimination with respect to worker welfare still occurs (see Theorem 1). On the other hand, it means that even a minority worker with many reviews and average rating score of 1 will face discrimination. However, this result comes in contrast to the empirical findings in online platforms [8, 1] which point towards belief-based discrimination. For example, as we have already mentioned in Section 1, Cui et al. [8] find that the existence of a review on the guest's profile can help attenuate discrimination on Airbnb; good reviews have the greatest impact but even a negative or a blank review can have positive effect on discrimination. In contrast, if social bias is belief-based, the employer behavior in our main model is consistent with the empirical observations. As Lemma 1 and Lemma 2 suggest, the difference in hiring probabilities of equally skilled minority and majority workers is amplified for workers with fewer or zero reviews. In the limit $K \to \infty$, where the number of reviews goes to infinity and uncertainty fades, hiring inequalities diminish (Theorem 2).

**Other representations of taste-based social bias.** Several other representations of taste-based social bias may be considered. One alternative model of taste-based social bias could be the following. Suppose that we modify the baseline model in Section 2 so that discriminating employers have the correct prior belief $q_0$ about minority workers' skill level but they knowingly discount their belief by $\beta$, i.e.

$$q_k^D = \beta \frac{G_k + G_0}{G_k + B_k + N_0}. \tag{E.2}$$

In this case, the difference $q_k^N - q_k^D = (1 - \beta)\frac{G_k + G_0}{G_k + B_k + N_0}$ is positive, but, in contrast to Lemma 1, does not necessarily decrease over time. On the contrary, the difference increases as the number of good reviews $G_k$ increases but reduces as the number of bad reviews $B_k$ increases. Hence, if social bias is taste-based, $N$ employers are mostly discriminating against minority workers with high rating scores. As in the previous model of taste-based social bias, such a behavior by discriminating employers might be less natural but is in fact inconsistent with the relevant empirical studies (see e.g. [8]). In any case, we note that a similar result to Theorem 1 also holds in this case of taste-based social bias.

Finally, a third alternative of taste-based social bias could be the following. Discriminating employers have the correct prior belief $q_0$ about minority workers but with some probability $p$ (per period) do not hire any minority worker regardless of the worker's review statistics. Similarly to the other taste-based models, the inequality in hiring opportunities still persists.

**DM policy under taste-based social bias.** It is important to mention that our DM policy (and its basic learning component) will also be effective for employers with taste-based social bias. Of course, the decrease in the discrimination gap may differ but the proof of Theorem 3 will follow along the same technical arguments. Hence, this result indicates that, given any of the aforementioned taste-based models or our belief-based model in Section 2, the DM policy will reduce the discrimination gap.

### E.2 Bayesian agents

In this paper, we assumed that agents are characterized by bounded rationality and limited computational ability in order to approximate a more realistic representation of real human behavior. However, studying an alternative model of Bayesian agents (with biased private signals or biased prior beliefs) would be an interesting direction to pursue.

Given our framework, Bayesian $D$ employers have misspeficied prior beliefs about minority workers. Instead of the naive rule (2), they use Bayes' rule to compute their belief

$$q_k^e = \mathbb{P}(Q = H \mid c, B_k, G_k)$$

that a worker of social group $c$ and review statistics $B_k, G_k$ is high-skilled. For simplicity, we can assume that employers also know $k$; alternatively, we could assume that they have a uniform prior on this variable (as in [2] and [10]). If employers ignore the existence of other types of employers other than their own, we claim that there exists discrimination against minority workers.

However, in more general settings, it is not clear whether discrimination can possibly change direction. For example, Bohren et al. [7] consider the case of Gaussian distributions in a - different from ours - two-period setting where some biased agents may have a lower, misspecified prior. Interestingly, they find that, under certain condition, the belief of the biased agents about the minority group may become larger than the belief of the impartial agents about the same group. Within a simplified version of our framework, it is possible to show that under more general conditions on the distribution of idiosyncratic preferences and for workers with at least one review, discriminating employers may have a higher belief for minority workers than majority workers. Exploring the exact relation of Bayesian agents and discrimination and comparing it to the case of naive agents is left for future research.

## Footnotes

[1] For technical completeness, in the case where $K \to \infty$, we can assume that $\lambda_i = \lambda_i(K) \to 0$, thus arrival rates scale down so that the total mass of agents in the market remains fixed at each time step (see also [11]).

[2] Note that $\Omega_k$ can be of any length 0 and $K - 1$.

[3]The *Kullback-Leibler divergence* between Bernoulli($p$) and Bernoulli($q$) is defined as $p \log(\frac{p}{q}) + (1 - p) \log(\frac{1-p}{1-q})$.

[4]Observe that, given our previous definitions, it holds that $\mathcal{H}_D = \mathcal{H}_D^+ = \mathcal{H}_N^-$.

[5]Several papers [4, 7, 8] model taste-based preference by adding a constant taste parameter to the agents' utility structure.

[6]That is, $\bar{a} + \mu_P - r \geq 0$, $\underline{a} + \mu_P + q_0 - r \geq 0$ as well as $\underline{a} + \bar{p} - r > 0$, $\bar{a} + 1 + \underline{p} - r < 0$ (see Assumption 1).