[Reviews · NeurIPS 2019]

Reviewer 1



As mentioned, the paper sets out to model labor markets in which employers are either discriminating (i.e., hold misspecified prior beliefs about a minority group) or non-discriminating, workers are either high skill or low skill, and employers generate reviews of workers. The paper further models hiring decisions that take into account a mix of prior beliefs and reviews, exploring the implications of these dynamics as more reviews come in and employers' assessment changes as a consequence. The paper demonstrates that such markets will end up generating less welfare for the minority group, even as employers learn over time. I expect my more expert colleagues will have a lot to say about this set up and the proofs. I found the set up reasonable and the findings rather instructive, but lack the technical expertise to evaluate them in depth. I do, however, feel well equipped to assess the proposed mechanism. To minimize the effect on worker welfare, the mechanism aims to learn whether employers are discriminatory, given their response to workers, and then avoid matching minority workers to these employers. If employers are understood to be interchangeable from the perspective of the worker (i.e., the worker would be equally happy to accept jobs from any of these employers), the proposed mechanism can mitigate the effects on minority workers' welfare. But employers and jobs are rarely so interchangeable. A good deal of discrimination in labor markets arises from the belief that members of certain groups (e.g., women) are not suitable for certain jobs (e.g., executive positions). In the real world, the discriminating employers are not randomly distributed by occupation or role. A mechanism that aims to avoid pairing a minority worker with a discriminating employer would only be effective -- and justifiable -- if all jobs are equally attractive from the perspective of the worker. I'm not convinced that a platform would be able to develop a taxonomy of job types such that any job in a type is essentially interchangeable.

Reviewer 2



This paper studies discrimination against minority workers in a two-sided market where an online platform matches employers and workers (e.g. ride-sharing). The proposed fairness goal is to equalize the expected 'welfare' of minority group workers with majority group workers. To model this two-sided market, they make the following assumptions: - There are two worker groups: the "minority" and the "majority." (Minority does not imply numerical minority, see comment in Improvements.) Both groups have the same percentage of "highly skilled" workers. - There are two employer groups: "discriminating" and "non-disciminating". The discriminating employers have a too-low prior about the probability that a minority worker is highly skilled. - The platform matches workers and employers in sequence with discrete timesteps. At each timestep, employers decide whether or not to hire the matched worker based on their estimate of the probability that the worker is highly skilled. This model does not allow employers to compare multiple workers before making the hiring decision in a given timestep. - To estimate the probability that a worker is highly skilled, the employers weigh their prior against the cumulative number of reviews that the worker receives, where a review is more likely to be poghedfgsfg6sitive if the worker is revealed to be highly skilled. Whether or not an employer is "discriminating" only affects their initial hiring decision and not their review. Originality: This paper's modeling isn't significantly more descriptive than what has been done in related work (e.g. Johari et al). The main difference from Johari et al. appears to be the modeling of employer history to determine whether or not an employer is discriminatory. This allows them to propose the "Direct Matching" intervention which would not have been possible under Johari et al.'s framework alone. Quality: Their theoretical results are straightforward and generally appear correct (with some minor issues in notation and Lemma 1, see Improvements). However, given that the theoretical results aren't particularly new or practically applicable, I think this paper could benefit from some empirical results (see Improvements). Clarity: The paper is well written overall, and the ideas presented are clear and well organized. The assumptions made by the model are outlined very clearly. My main concern is in the notation -- the notation is not always precise in its treatment of random variables vs. the values those random variables take (for example, Equation (3), see Improvements). Significance: The model of discrimination in a two-sided market is especially salient with the gig economy becoming more and more economically relevant. However, their model falls short of applying to real-world review based platforms in a few major ways, including assuming that discrimination only affects the hiring decision and not the reviews (see Improvements), and assuming that employers do not compare multiple workers before making their hiring decision at a given timestep, which happens on many review-based hiring platforms.

Reviewer 3



Summary: They provide a framework to think about discrimination in a setting where employers are matched to employees. Each employee is either in the majority or minority group and each employee is either qualified or unqualified. And each employer is either non-discriminatory or discriminatory. The discrimination from the employers come from a wrong prior belief about the minority group. Each employee has a history of reviews (good or bad and employers may not necessarily leave a review). Because discrimination arises because of a wrong prior belief about the minority group, as long as an employee has good reviews, each employer will use the reviews instead of their prior beliefs. They propose an algorithm that reduces the discrimination faced by minority employees. First, for some amount of time, the system uniformly at random matches employers with employees. Then based on the review history, employers are labelled as discriminatory, non-discriminatory, or in the case when an employers has not left enough reviews, they are not known to be either discriminatory or non-discriminatory. Then minority workers are essentially matched with the non-discriminatory employees whereas the majority workers are matched with the discriminatory employees. Their algorithm is contrasted with the algorithm that just assigns workers uniformly at random to employees. The welfare of the majority class is the same under either algorithm. However, eventually the welfare of minority employees increases under their algorithm versus the baseline algorithm. In terms of their theoretical contributions: They provide a theoretical analysis of the equilibrium of the dynamical system showing that at equilibrium works from a minority class on average have lower welfare than workers from the majority class; they provide a theoretical analysis showing that eventually the fraction of good reviews (potentially adjusted by the discriminatory employer's prior beliefs) of qualified (respectively unqualified) workers converge to the same number regardless of minority or majority status. They also provide a theoretical justification for why an algorithm they propose reduces discrimination. -------------- Originality: As far as I am aware, the framework proposed in this paper is original. What makes their framework unique from other works that study discrimination in labor markets is incorporation of reviews on the employees. However, I am not completely familiar with this area of research so I could be mistaken. The theorems appear to be novel and the tools used to establish these results are drawn on from other works (see "Techniques" section). ----------- Quality: I think that the quality of this work is good overall. This is a theoretical paper and the theorems seem to be technically sound. However, they could have included synthetic experiments to illustrate their theory. They cover a lot of material (the framework, theory on the properties of the framework, and then an algorithm to reduce discrimination) in this paper, but I do think that it tells a complete story and removing any of these pieces wouldn't make sense. To add to the story, however, it would be nice to know how long / how many samples it takes for the welfare of the minority group to begin to improve according to their algorithm. The only result on their algorithm is that eventually the welfare of the minority group improves compared to the minority group's welfare under the algorithm that matches employees and employers uniformly at random. The framework is a bit simplistic (they don't address algorithmic bias in terms of feedback loops in the review system) but to study discrimination in these types of systems, I think their paper provides a fine starting point. ----------- Clarity: Although a lot of notation has to be introduced, I think that this paper was clear overall. They provide high level explanations of each of their theorems as well as insights into the proofs/tools used. However, there are some parts of the paper that were unclear to me. 1) Perhaps I missed it but, it is unclear to me how to compute equation 10, i.e. what is P(g = D| H_n)? 2) Another area that is a bit confusing to me is Section 4.2. I understand how the employers are in the learning pool because we are trying to figure out which employers are discriminatory. However, why are there workers in the learning pool? Are these just workers that cannot be matched to anyone out of the matching pool? Also in line 305, why are the worker that have been matched with N-identified employers prioritized? What difference does it make if they have been matched before or not with N-employers? Minor comment to the authors: - The line after eqn (1): I am not familiar with what "ex ante idiosyncratic" and "ex post idiosyncratic" means, so it would be good to provide a reference to these terms or briefly define them. ---------- Significance: I think this work is fairly significant. They provide a new framework for thinking about an important problem (discrimination in systems like Uber and Airbnb) with many possible extensions in the future. As they acknowledge, their framework is limited. For instance, their framework does not encompass what happens when there are feedback loops in the review system, i.e. if someone starts off with bad reviews even though they are a qualified worker, these initial bad reviews can beget even more negative reviews. Their model essentially assumes that as long as someone is given a chance at being hired, eventually the proportion of good reviews converges to some number depending on the qualification of that person regardless of majority/minority status (and they cite empirical work that supports this framework). Except for legal concerns, the algorithm they proposed could potentially be used in real world systems. I think that the results of most of the theorems are not surprising as the authors also remark. However, going from the framework to the theory is non-trivial and necessary.

[Author Response · NeurIPS 2019]

We thank the reviewers for their thoughtful comments. Regarding the empirical evaluation of our policy, we plan to
include simulations to showcase the efficiency of the policy; due to lack of time we leave this for the final version of the
paper. Next, we address each reviewer's additional questions separately.
**Review 1:** — Regarding non-interchangeable occupations and jobs: Thank you for raising your concern; this is an
interesting aspect. We currently assume that there is one fixed job type and the same distribution of high-skilled workers
across social groups. With multiple job types and belief-based bias, we could again improve the welfare of minority.
Specifically, in order to learn whether an employer is discriminating (see eq. (10)), we need to condition on the job type
$r$ and the bias level $\beta_r$ of discriminating employers against minority workers doing job $r$, as well as take into account
the fraction of $D$ employers looking for task $r$. Thus, we think that the DM policy can be modified to apply here, with
the difference that matching decisions will also depend on job types and capacity constraints. Our results will still hold
even if workers stick to their type of jobs as long as workers do accept offers from many different employers of the
same type (who offer the same job). This is common in many types of jobs in online labor platforms.
— Regarding the concern about reinforcing unjustified stereotypes: Given belief-based bias, our policy reduces the
effect of unjustified stereotypes. It achieves it by helping minority workers accumulate a larger number of reviews. As
the available information about the worker increases, the effect of stereotypes (belief-based bias) reduces. Theoretically,
bias can not be reinforced asymptotically (Theorem 2). But even in a practical setting with a finite time horizon, our
policy helps decrease faster the uncertainty about minority workers; as a result, the discrimination gap also decreases.
— Thank you for the references. Indeed, studying the exit rate of workers is an interesting direction. Intuitively, we
expect higher exit rates of minority workers because "they may not even have the chance to receive enough reviews or
even stay long in the platform due to the competition" (l. 241-243). Hence our DM policy may help to reduce exit.
— Regarding the applicability of DM policy: There is no available law that clearly regulates such policies (see Rosenblat
et al. (2017), Levy and Barocas (2017)). However, platforms are already allowed to collect data about users to optimize
the platform's actions such as matching and recommendations. Policies such as hiding sensitive information (see the
Airbnb policy about user photos) have not been successful. Furthermore, policies that penalize discriminating users may
be problematic because 1) under belief-based bias, it is not clear how to select a threshold for discriminatory behavior,
and 2) such policies may create imbalance in the market. Thus, our policy could be an effective, easy alternative to
implement and/or serve as a benchmark for future policies. Nevertheless, online platforms already implement similar
directed matching (e.g. new users with few reviews are more frequently shown on top of search results).
**Review 2:** — In comparison to Johari et al. (2017), we include agent histories on both sides of the market and
incorporate strategic behaviors (social learning and hiring/review decisions) on one side of the market (employers) to
the evolution of the system (lines 83-84). These two factors make the dynamical system in our paper take a non-linear,
non-standard form, and differentiate (both technically and conceptually) our model from Johari et al. (2017).
— Regarding bias in reviews: We could directly extend the model to add an additional bias level for reviews. In this
case, the model exhibits taste-based discrimination (and not belief-based as we mainly consider). As we discuss in
Appendix E, given taste-based bias, discrimination persists asymptotically but our DM policy can still improve the
welfare of minority workers. The same would hold if we assume additional bias in reviews. In both cases of belief- and
taste-based bias, DM helps minority workers accumulate a larger number of reviews faster.
— Regarding comparison of multiple workers: Modeling this will complicate the model by introducing choice models.
However, if we borrow ideas from search theory, then the problem of each employer reduces to a threshold-based
decision rule which - in expectation - should not affect our current results.
— On notation in eq. (3): Indeed, we mean $W_q^c(K) = \mathbb{E}(\sum_{k=1}^{K} \delta^k m_k \mid Q = q, C = c)$. Thank you for the correction.
— On use of "monotonic": We actually mean non-strictly increasing/decreasing.
— On word choice (minority/majority): Our results do not rely on assumptions about the size of each worker group.
Thus, priviledged/unpriviledged is a better choice and we will adopt it in the paper.
**Review 4:** — Thank you for acknowledging the originality of our model. We also view this as a significant contribution.
— $\mathbb{P}(g = D \mid H_n)$ is the probability that the employer belongs to group $D$ (discriminating) given her history $H_n$ of past
hiring decisions about the $n-1$ workers she has met so far.
— "Ex-ante/ ex-post idiosyncratic": We mean the preference shocks of employers before/ after hiring the worker,
respectively (see also Besbes and Scarsini 2018). We will include the definition.
— On the optimality of DM policy: The optimality of DM policy is an open, challenging question. However, the policy
is simple and we show that it successfully reduces discrimination.
— On Section 4.2 and the learning pool: Some workers are in the learning pool, because we are learning from employers'
decisions made for those workers. Under the given DM policy, some minority workers enter the learning pool and a
few may remain there until they leave. An alternative solution (which does not affect our technical analysis) would be
to carefully randomize among workers of the same history so that minority workers in the learning pool also exit the
learning pool with positive probability.

[Meta-Review · NeurIPS 2019]

This is an interesting paper which deals with the setting of matching employees to employers that may be potentially biased. The results are exciting and of interest to the NeurIPS community. Thus I recommend acceptance of the paper. The paper should, however, make the relation to the closely related work [27] very clear in the camera ready version (see review 2), instead of having just one sentence hidden in "techniques".